# Surveillance versus Adjuvant Treatment with Chemotherapy or Radiotherapy for Stage I Seminoma: A Systematic Review and Meta-Analysis According to EAU COVID-19 Recommendations

**DOI:** 10.3390/medicina58111514

**Published:** 2022-10-24

**Authors:** Dong Hyuk Kang, Kang Su Cho, Jae Yong Jeong, Young Joon Moon, Doo Yong Chung, Hae Do Jung, Joo Yong Lee

**Affiliations:** 1Department of Urology, Inha University College of Medicine, Incheon 22212, Korea; 2Department of Urology, Gangnam Severance Hospital, Urological Science Institute, Yonsei University College of Medicine, Seoul 06273, Korea; 3Department of Urology, Severance Hospital, Urological Science Institute, Yonsei University College of Medicine, Seoul 03722, Korea; 4Department of Urology, Inje University Ilsan Paik Hospital, Inje University College of Medicine, Goyang 10380, Korea; 5Center of Evidence Based Medicine, Institute of Convergence Science, Yonsei University, Seoul 03722, Korea

**Keywords:** testis cancer, seminoma, active surveillance, COVID-19

## Abstract

*Background and Objectives*: During the coronavirus disease 2019 (COVID-19) outbreak, the European Association of Urology (EAU) Guidelines Office Rapid Reaction Group (GORRG) recommended that patients with clinical stage I (CSI) seminoma be offered active surveillance (AS). This meta-analysis aimed to evaluate the efficacy of AS versus adjuvant treatment with chemotherapy or radiotherapy for improving the overall survival (OS) of CSI seminoma patients. *Materials and Methods*: A systematic review was performed according to Preferred Reporting Items for Systematic Reviews and Meta-Analyses guidelines. The PubMed/Medline, EMBASE, and Cochrane Library databases were searched. The primary outcome was 5-year OS, and the secondary outcome was the 5-year relapse-free survival (RFS). The outcomes were analyzed as odds ratios (ORs) and 95% confidence intervals (CIs). *Results*: A total of 14 studies were included. Overall, the quality scores were relatively high, and little publication bias was noted. In terms of the 5-year OS, 7 studies were analyzed; there was no significant difference between AS and adjuvant treatment (OR, 0.99; 95% CI, 0.41–2.39; *p* = 0.97). In terms of 5-year RFS, 12 studies were analyzed. Adjuvant treatment reduced the risk of 5-year recurrence by 85% compared with AS (OR, 0.15; 95% CI, 0.08–0.26; *p* < 0.001). *Conclusions*: In terms of the OS in CSI seminoma patients, no intergroup difference was noted, so it is reasonable to offer AS, as recommended by the EAU GORRG until the end of the COVID-19 pandemic. However, since there is a large intergroup difference in the recurrence rate, further research on the long-term (>5 years) outcomes is warranted.

## 1. Introduction

Testicular germ cell tumors are the most common carcinomas in young men, accounting for approximately 1% of all carcinomas, and these are divided into seminoma and non-seminoma [1]. About 50% are seminomas and 80–85% are stage I disease, so seminoma clinical stage I (CSI) is the most common clinical condition [2]. Despite major changes in the management of CSI seminoma in the past 20 years, very few reports of CSI seminoma have been published over last decade due to its low incidence.

In the past, adjuvant radiotherapy (RT) after radical orchiectomy was the general treatment with a low recurrence rate [3], but with the advent of cisplatin-based chemotherapy (CT), the late side effects of RT [4], and further developments in diagnostic technology, active surveillance (AS) has become more common. Therefore, AS has been accepted as the standard for more than 10 years, and many guidelines recommend it as the preferred option for CSI seminoma [5,6]. Although AS is reportedly associated with a higher recurrence rate than adjuvant therapy, the treatment of testicular cancer recurrence shows a very high success rate even in the salvage treatment setting, which is an advantage of AS [7]. Nevertheless, several concerns remain. First, AS can avoid treatment-related toxicity problems by avoiding unnecessary treatment, but it can lead to an undue burden of overall treatment. In addition, in cases of recurrence, the need for various treatment methods may increase [8]. Finally, AS requires repeated imaging examinations for at least 5 years. Therefore, AS is not the only option, and adjuvant therapy can bring greater benefits to selected patients.

The coronavirus disease 2019 (COVID-19) pandemic has created a global health threat. This disease has effected major change in the priorities of medical and surgical procedures. In the context of the COVID-19 pandemic, the European Association of Urology (EAU) Guidelines Office Rapid Reaction Group (GORRG) presented a response policy for diagnosis, treatment, and follow-up, according to the priority of the urologic disease [9]. Guidelines for testicular cancer management recommended that patients with seminoma and low-risk non–germ cell tumor clinical stage I (CSI) be offered AS as the first choice of management for CSI testicular cancer during COVID-19. Therefore, it is important to compare AS and adjuvant treatments through a meta-analysis according to the EAU COVID-19 recommendation. Although previous meta-analyses have been published [10], it would be beneficial to verify the effectiveness of AS by AS and adjuvant treatment during the COVID-19 pandemic by analyzing more cases.

## 2. Materials and Methods

### 2.1. Literature Search and Study Selection

The meta-analysis and report of the results were based on the PRISMA checklist and the details are shown in Appendix A. A comprehensive literature search of the PubMed Central, Cochrane Central Controlled Register of Trials (CENTRAL), and EMBASE databases was performed to search for articles published until 31 October 2021. The potential gray literature, relevant conference abstracts, and reference list of identified studies were also searched. We used search terms such as “seminoma”, “chemotherapy”, “radiotherapy”, “surveillance”, “observation”, and combinations of these search terms. Titles and abstracts were first screened for relevance, and papers meeting the inclusion criteria were subjected to full-text screening. All duplicate articles were excluded, and the reference lists of the retrieved articles were manually searched to identify any other relevant studies. The detailed methods we used to search literatures are shown in Appendix A. This meta-analysis was registered in PROSPERO (number: CRD42021299235).

### 2.2. Inclusion and Exclusion Criteria

This systematic review and meta-analysis followed the participants, interventions, comparators, outcomes, and study design (PICOS) and Preferred Reporting Items for Systematic Reviews and Meta-Analyses guidelines. The PICOS model was used to construct and answer the following clinical criteria: (1) Patients: patients with CSI seminoma; (2) Intervention: adjuvant CT or RT for CSI seminoma; (3) Comparison: cases wherein AS was implemented for CSI seminoma; (4) Outcome: 5-year overall survival (OS) and 5-year relapse-free survival (RFS); and (5) Study type: randomized clinical trials and prospective and retrospective studies. The following inclusion criteria were adopted: (1) retrospective and prospective studies reporting data on any adjuvant CT or RT versus AS alone for patients with CSI seminoma; (2) availability event or rate of relapses and/or deaths within 5 years; and (3) English publication. The exclusion criteria were reporting of duplicate publications, lack of data on 5-year outcomes, and study populations of mixed seminoma and non-seminoma patients.

### 2.3. Data Extraction and Quality Assessment

All of the retrieved articles were independently screened by two reviewers (D.H.K. and H.D.J.), while two other reviewers (D.Y.C. and Y.J.M.) independently analyzed all details of each article to confirm that each met the inclusion criteria. Any discrepancy in the opinion of the two reviewers was resolved through discussion until consensus was reached or via third-party adjudication performed by another reviewer (J.Y.L.). Once the final group of articles was selected, two researchers independently examined study quality. The quality of each study was estimated using the Newcastle–Ottawa Scale (NOS) for non-randomized controlled trials (www.ohri.ca/programs/clinical_epidemiology/oxford.asp, accessed on 15 March 2022) [11]. The maximum NOS score was nine points. A total score of less than or equal to 5 points was considered low, 6–7 was considered intermediate, and 8–9 was considered high quality.

### 2.4. Data Synthesis and Analysis

The primary outcome measure was 5-year OS, while the secondary outcome measure was 5-year RFS. The meta-analysis was conducted using R software (version 4.1.2, R Foundation for Statistical Computing, Vienna, Austria; http://www.r-project.org, accessed on 1 July 2022). The effect of the measures of the outcomes of interest were the odds ratios (ORs) with 95% confidence intervals (CIs) obtained by extracting the proportion of patients with outcome events. Pooled incidences were calculated using a fixed-effects or random-effects model according to the heterogeneity of the included studies. If no significant statistical inconsistency was observed, the summary estimate was calculated using the fixed-effects model. When heterogeneity was observed, the summary estimate was calculated using a random-effects model.

### 2.5. Ethics Statement

The data and results used in this paper are all from published studies, and there is no ethical issue, so the approval of the ethics committee is not required.

## 3. Results

### 3.1. Study Characteristics

The database search identified 385 articles that could be included in this meta-analysis. Of these, 335 articles were excluded based on our selection criteria after the titles and abstracts were screened. The remaining 50 studies evaluated OS and RFS. Following a full-text review, 36 studies were excluded because they reported irrelevant results. Figure 1 shows the results of the comprehensive search, selection process, and number of excluded studies along with the relevant evidence. The search identified 14 studies with a total of 12,358 patients (Table 1) [12,13,14,15,16,17,18,19,20,21,22,23,24,25].

### 3.2. Quality Assessment

The results of the quality assessment based on the NOS are shown in Table 1. The scores of all studies were 6 points or higher, indicating relatively high quality.

### 3.3. Publication Bias

Funnel plots from these meta-analyses are shown in Figure 2. For 5-year OS and RFS, a little publication bias was demonstrated on the funnel plots.

### 3.4. Heterogeneity Assessment

Forest plots for the 5-year OS and RFS are shown in Figure 3 and Figure 4. There was high heterogeneity for 5-year OS (I^2^ = 72%, *p* = 0.006) and 5-year RFS (I^2^ = 71%, *p* < 0.001); therefore, a random-effects model was used. After selection of the effect models, little heterogeneity was observed in the L’Abbe plots (Figure 5) and radial plots (Figure 6). We conducted sensitivity analysis for the outcome reporting bias (ORB) to examine the degree of heterogeneity (Figure 7). The sensitivity of this meta-analysis was considered robust, as the results on 5-year OS and 5-year RFS were not affected until up to five studies were excluded.

### 3.5. Forest Plot Results

#### 3.5.1. Five-Year OS

Seven trials were included in this analysis (N = 9867 patients). There was no significant difference between AS and adjuvant therapy in 5-year OS (OR, 0.99; 95% CI, 0.41–2.39; *p* = 0.97) (Figure 3). The 5-year mortality rates in the adjuvant therapy and AS groups were 2.0% and 2.6%, respectively.

#### 3.5.2. Five-Year RFS

Twelve trials were included in this analysis (N = 4806 patients). Adjuvant RT or CT reduced the risk of 5-year recurrence by 85%, compared with AS (OR, 0.15; 95% CI, 0.08–0.26; *p* < 0.001) (Figure 4). The 5-year relapse rates were 3.7% and 15.8% in the adjuvant therapy and AS groups, respectively.

## 4. Discussion

COVID-19 is an acute respiratory infectious disease caused by severe acute respiratory syndrome coronavirus 2 (SARS-CoV-2), a novel strain of coronavirus that was first reported in November 2019 [26]. The rapid spread of COVID-19 has had a dramatic impact on individuals and medical systems globally. In addition to people infected with SARS-CoV-2, intense demand on the limited medical system resources has contributed to the spread of the virus, including reduced capacity and rapid depletion of medical systems and hospitals [27]. To prevent viral spread, it is increasingly recommended that hospital visits be limited by delaying surgery or reducing unnecessary treatments [28]. However, medical delays or avoidance can increase morbidity and the risk of death associated with treatable and preventable health conditions as well as contribute to excess reported deaths directly or indirectly related to COVID-19. Due to the novelty of this situation, it is difficult to determine gains and losses through treatment delays or avoidance. Therefore, it is very important to establish individual guidelines for the management of each disease.

Although COVID-19 has a tremendous impact in the field of urology, many urologists are making efforts to provide patients with the best treatment for the current situation. In response to the COVID-19 pandemic, the EAU Guidelines Office has been working with the Executive Committee, the Section offices, and others to set up a Rapid Reaction Group [9,29,30]. The protocol was divided into four large umbrellas of recommendations: diagnosis, surgical treatment and medical therapy, follow-up/telemedicine, and emergency treatment. The panel provides a table with recommendations according to priority level. They created four color-separated risk stratification tools to help apply recommendations: low priority: clinical harm very unlikely if service postponed for 6 months (green color); intermediate priority: cancel but reconsider in case of increase in capacity, clinical harm possible if postponed for 3 months but unlikely (yellow color); high priority: the last to cancel, prevent a delay >6 weeks, and clinical harm very likely if postponed for >6 weeks (red color); and emergency: cannot be postponed for >24 h due to a life or organ–functionthreatening condition (black color).

All urological cancers, including testicular cancer, were evaluated by the EAU GORRG. The EAU Guidelines Office recommended that patients with seminoma and low-risk non-seminomatous germ cell tumor (NSGCT) CSI be offered AS as the first choice of management in CSI testicular cancer during COVID-19 (green color). Although AS is usually recommended for seminoma CSI, adjuvant treatment also features sufficient benefits in some cases, especially in terms of reducing the recurrence rate [7,31,32]. We concluded that a meta-analysis of AS versus adjuvant treatment was necessary. Although a meta-analysis was published in 2015 [10], the reinterpretation of data with the addition of studies published since then in the current COVID-19 situation over several years, is valuable. The present meta-analysis is similar to the previous meta-analysis [10], but a high-quality study has been added, and the data of all the studies included in the meta-analysis have been re-evaluated in detail to correct the errors. In addition, a validation of quality assessment, publication bias, and heterogeneity assessment, which had not been conducted in previous meta-analyses, has been conducted, is different from the previous study, and is considered to be of higher value.

The present systematic review and meta-analysis is the first to address this argument in the current COVID-19 epidemic. Here, we found no difference in 5-year survival rates between patients treated with AS versus adjuvant treatment. In terms of OS, we believe that implementing AS in the current COVID-19 pandemic is reasonable. However, since the 5-year RFS showed better results than implementing AS as adjuvant treatment, further discussion is needed to determine the best course of action. First, it should be recognized that seminoma CSI shows a notable treatment success rate even in cases of recurrence. Second, there is an unnecessary medical burden due to adjuvant treatment. And lastly, there is a possibility that side effects may occur due to adjuvant treatment. Therefore, even if the disease recurs in the future, it is more reasonable to consider adjuvant treatment as unreasonable considering the reduction in the risk of exposure to COVID-19 and medical ability by implementing adjuvant treatment.

One concern remains regarding AS in patients with seminoma CSI. The question is whether more intensive treatment is needed in the event of recurrence than in cases in which adjuvant therapy was used. This problem is called the “treatment burden”. One study investigated 164 patients receiving AS, of whom 22 (13%) experienced recurrence [8]. Of them, 6 of 13 patients treated with RT and 2 of 9 patients treated with CT for recurrence experienced a second recurrence (total second recurrence rate, 36%). In contrast, another study showed that the proportions of patients requiring CT were similar in the AS and adjuvant RT groups [20]. Therefore, when AS was used as an initial management strategy, the initial treatment burden for patients was reduced, while the follow-up treatment burden was not greater than that of patients in the adjuvant RT group. Therefore, our study is considered to be very valuable because this study investigated the effectiveness of AS to reduce the burden of medical care during the COVID-19 period.

This study has several limitations. First, most of the included studies were retrospective and only two were prospective. It also includes a series of reporting data from decades ago when RT technology was different. The 5-year event OR can only be extrapolated or calculated from the published data. Our analysis included public data and not individual patient data. In addition, since there were few clinical studies on NSGCT, they were not included in our analyses. Therefore, it was impossible to verify NSGCT according to the guidelines presented by the EAU. In this regard, more clinical studies on NSGCT will be needed to verify NSGCT. Nevertheless, this study can have great significance as it is the first meta-analysis to validate the EAU guideline for testicular seminoma CSI in the COVID-19 pandemic. A randomized control trial is needed to validate our findings. In addition, since most testicular cancer patients are young, a study of long-term outcomes (longer than 5 years) must be conducted.

## 5. Conclusions

In terms of the OS in CSI seminoma patients, we found no intergroup difference; thus, it is reasonable to offer AS as recommended by the EAU GORRG until the end of the COVID-19 pandemic. However, since we noted a large intergroup difference in recurrence rates, more research is needed regarding long-term (>5 years) outcomes.

## Figures and Tables

**Figure 1 medicina-58-01514-f001:**
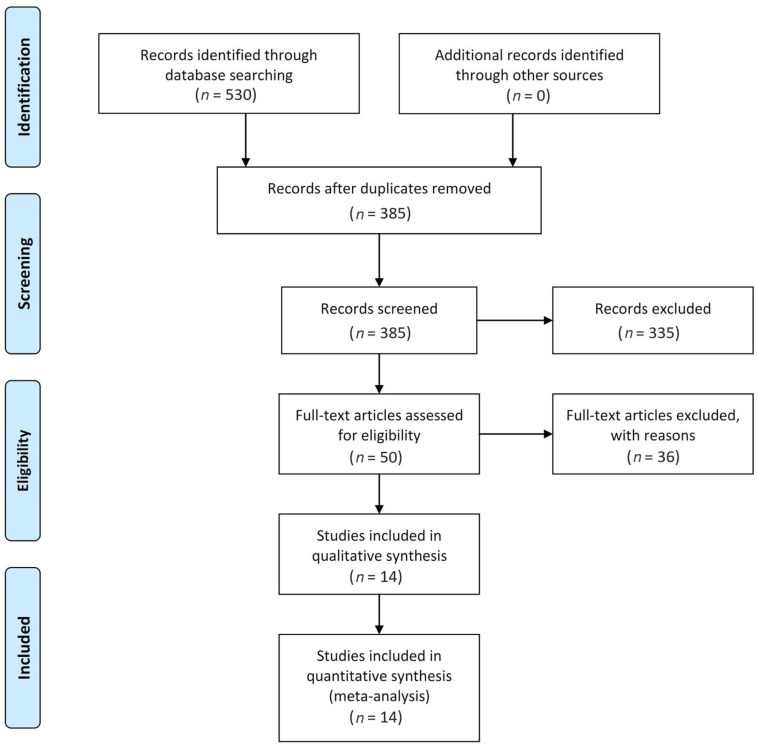
Search strategy for the systematic review and meta-analysis.

**Figure 2 medicina-58-01514-f002:**
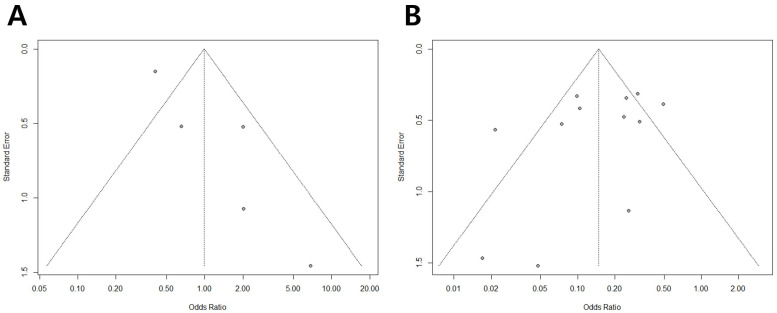
Funnel plots of 5-year overall survival (OS) (**A**) and 5-year recurrence-free survival (RFS) (**B**). Some publication bias was noted in the funnel plots.

**Figure 3 medicina-58-01514-f003:**
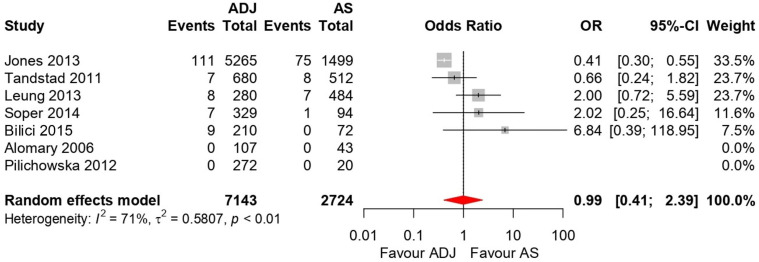
Forest plots for 5-year overall survival (OS) comparing active surveillance (AS) and adjuvant therapy [12,14,16,20,21,23,24]. There was no significant difference between AS and adjuvant therapy in 5-year OS (odds ratio, 0.99; 95% confidence interval, 0.41–2.39; *p* = 0.97).

**Figure 4 medicina-58-01514-f004:**
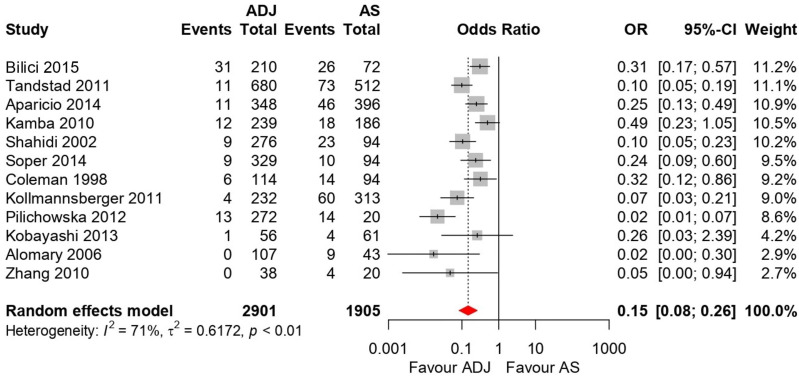
Forest plots for 5-year recurrence-free suvival (RFS) comparing active surveillance (AS) and adjuvant therapy [12,13,14,15,17,18,19,21,22,23,24,25]. Adjuvant radiotherapy or chemotherapy reduces the risk of 5-year recurrence by 85% versus AS (odds ratio, 0.15; 95% confidence interval, 0.08–0.26; *p* < 0.001).

**Figure 5 medicina-58-01514-f005:**
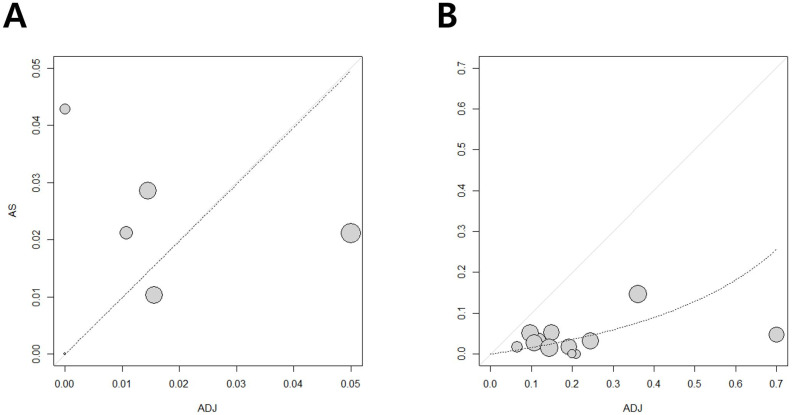
L’Abbe plots of 5-year overall survival (OS) (**A**) and 5-year recurrence-free survival (RFS) (**B**). A little heterogeneity was noted in the L’Abbe plots.

**Figure 6 medicina-58-01514-f006:**
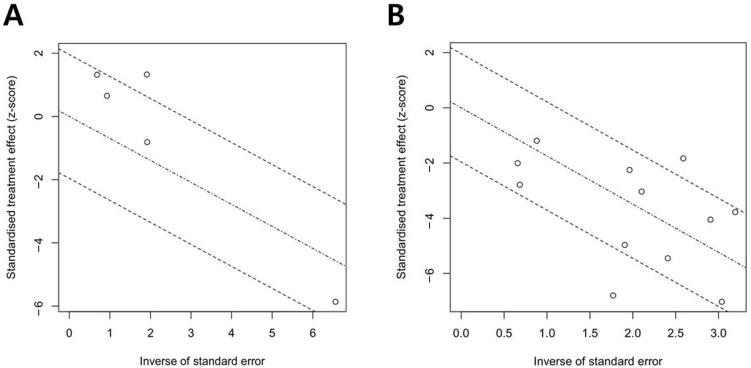
Radial plots of 5-year overall survival (OS) (**A**) and 5-year recurrence-free survival (RFS) (**B**). A little heterogeneity was noted in radial plots.

**Figure 7 medicina-58-01514-f007:**
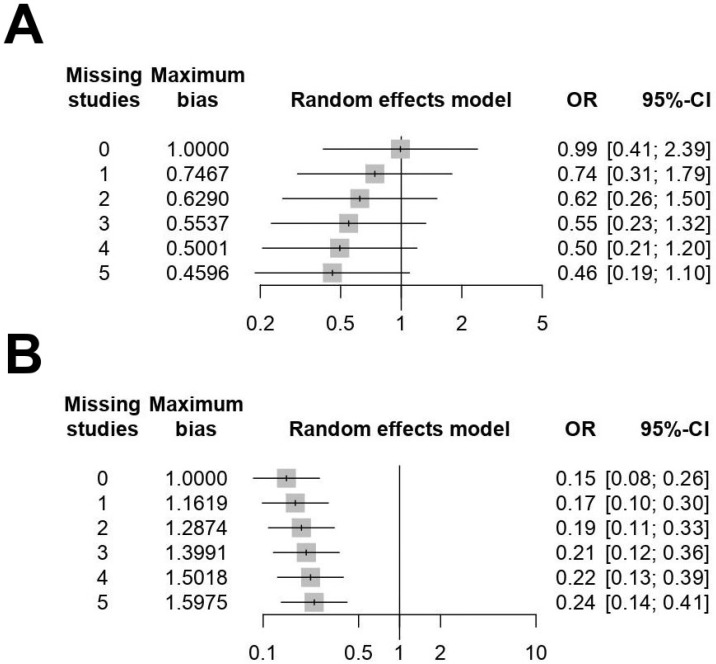
Sensitivity analysis for the outcome reporting bias (ORB) of 5-year overall survival (OS) (**A**) and 5-year recurrence-free survival (RFS) (**B**). The sensitivity was considered robust in the sensitivity analysis for ORB.

**Table 1 medicina-58-01514-t001:** Characteristics of the Included Studies.

Study	Year	Study Type	Number of Patients					5-year OS %		5-year RFS %		Median F/U Months	Quality Assessment
			Total	AS	ADJ			AS	ADJ	AS	ADJ		
					Total	CT	RT						
Coleman [15]	1998	Retrospective	239	94	144	-	144	NA	NA	85.1	95.8	NA	7
Shahidi [22]	2002	Retrospective	370	94	276	-	276	NA	NA	75.5	96.7	122	6
Alomary [12]	2006	Retrospective	150	43	107	-	107	100	100	79	100	54	6
Kamba [17]	2010	Retrospective	425	186	239	57	182	NA	NA	90	95	52.5	7
Zhang [25]	2010	Retrospective	50	20	30	30	-	NA	NA	80	100	NA	7
Kollmannsberger [19]	2011	Retrospective	545	313	232	73	159	NA	NA	80.7	98	33/33/65 (AS/CT/RT)	7
Tandstad [24]	2011	Prospective	1192	512	680	199	481	98.4	99	85.7	98.4	60/41/73 (AS/CT/RT)	8
Pilichowska [21]	2012	Retrospective	292	20	272	200	75	100	100	31.4	95.2	76.5	6
Jones [16]	2013	Retrospective	6764	1499	5265	-	5265	95	97.9	NA	NA	91	6
Kobayashi [18]	2013	Retrospective	118	61	56	-	56	NA	NA	93.4	98.2	67/174 (AS/RT)	7
Leung [20]	2013	Retrospective	764	484	280	-	280	98.6	97.2	NA	NA	79/102 (AS/RT)	8
Aparicio [13]	2014	Prospective	744	396	348	348	-	NA	NA	88.3	96.8	80	7
Soper [23]	2014	Retrospective	423	94	329	-	329	98.8	98	89.2	97.2	62/90 (AS/RT)	7
Bilici [14]	2015	Retrospective	282	72	210	80	130	100	95.7	64.2	93.8	38.5	7

ADJ, adjuvant treatment; AS, active surveillance; CT, chemotherapy; OS, overall survival; RFS, relapse-free survival; RT, radiotherapy.

## Data Availability

The data presented in this study are available in the article.

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
