# Peer review of "Surveillance versus Adjuvant Treatment with Chemotherapy or Radiotherapy for Stage I Seminoma: A Systematic Review and Meta-Analysis According to EAU COVID-19 Recommendations"

_medicina, 2022, doi:10.3390/medicina58111514_

Round 1

Reviewer 1 Report

This is a well done meta analysis investigating the specific question of surveillance versus adjuvant therapy in stage I seminoma. It confirms that surveillance and adjuvant therapy likely have similar OS, and that differences in RFS exist and often thus center decision making on treatment burden and burden of follow-up. The authors performed a good review of the state of evidence to date, especially given a stronger recommendation toward surveillance than previously offered. 

Author Response

This is a well done meta analysis investigating the specific question of surveillance versus adjuvant therapy in stage I seminoma. It confirms that surveillance and adjuvant therapy likely have similar OS, and that differences in RFS exist and often thus center decision making on treatment burden and burden of follow-up. The authors performed a good review of the state of evidence to date, especially given a stronger recommendation toward surveillance than previously offered.

Answer: Thank you very much for your review.

Reviewer 2 Report

Comments to the Author

The COVID-19 pandemic caused a major shift in priorities for medical and surgical procedures. Affecting care for oncological conditions, such as TGCT. Of this neoplasm, the CSI was one of the most affected stages due to the predominance of these sTGCT cases. Because of this, Dong Hyuk Kang and colleagues have addressed this important question by assessing the relevance of AS versus adjuvant treatment in these patients through meta-analysis, as recommended by the Association Guidelines Office Rapid Reaction Group. European Union of Urology (UAE).

The data presented in this study “Surveillance versus adjuvant treatment with chemotherapy or radiotherapy 2 for stage I seminoma: a systematic review and meta-analysis according to EAU COVID-19 recommendations”  have some shortcomings, as noted below.

1)    In general, this manuscript presents very similar results to those published by Petrelli et al., 2015. This confirms the relevance of AS and adjuvant treatment in patients with CSI testicular tumor, as well as their effect on disease recurrence. Situation that does not change in the context of the COVID19 pandemic. This fact is crucial to determine the relevance, originality and incidence of the results shown here.

2)    It should be noted that Dong Hyuk Kang interestingly highlights intergroup differences in recurrence rates, which could strengthen this study. Therefore, it is suggested to go deeper into this point, with the intention of differentiating this manuscript from what has been previously published.

3)    On the other hand, the discussion is very short, you should reinforce this section.

4)    Finally, there is a pagination mismatch due to the final point in figure 1.

5) In methodology section it is mentioned that "relevant variants" were contemplated, however, these are not reflected in the results, as if it is shown in what was published by Petrelli 2015.

Author Response

The COVID-19 pandemic caused a major shift in priorities for medical and surgical procedures. Affecting care for oncological conditions, such as TGCT. Of this neoplasm, the CSI was one of the most affected stages due to the predominance of these sTGCT cases. Because of this, Dong Hyuk Kang and colleagues have addressed this important question by assessing the relevance of AS versus adjuvant treatment in these patients through meta-analysis, as recommended by the Association Guidelines Office Rapid Reaction Group. European Union of Urology (EAU).

The data presented in this study “Surveillance versus adjuvant treatment with chemotherapy or radiotherapy 2 for stage I seminoma: a systematic review and meta-analysis according to EAU COVID-19 recommendations” have some shortcomings, as noted below.

Comment 1 In general, this manuscript presents very similar results to those published by Petrelli et al., 2015. This confirms the relevance of AS and adjuvant treatment in patients with CSI testicular tumor, as well as their effect on disease recurrence. Situation that does not change in the context of the COVID19 pandemic. This fact is crucial to determine the relevance, originality and incidence of the results shown here.

Answer 1 Thank you for your comment. We agree that Petrelli et al.'s study and our study are similar in some respects, but we also want to emphasize that there are differences. First, a study by Bilici et al. was added to this analysis. This study received a relatively high score in the quality assessment as well as a relatively high weight in the forest plots of OS and RFS, so it is considered a very important study in this meta-analysis. We also re-evaluated the data from all studies included in the meta-analysis in detail, and found that some studies needed correction of their results. In other words, it was confirmed that there were errors in the data in some of Petrelli et al.'s study, and our study published corrected results. There is a subtle difference in the results from Petrelli et al. Finally, the verification of quality assessment, publication bias, and heterogeneity assessment, which were not conducted in Petrelli et al., was conducted in this study. These are the most basic contents of the meta-analysis, and since this study has been verified, it can be said that there is a difference from the previous study. Contents about this have been added to 'Discussion'.

The present meta-analysis is similar to the previous meta-analysis [10], but a high-quality study has been added, and the data of all the studies included in the meta-analysis have been re-evaluated in detail to correct the errors. In addition, validation of quality assessment, publication bias, and heterogeneity assessment, which was not conducted in previous meta-analysis, has been conducted and is different from previous study and is considered to be of higher value. [Line 225-230]

Comment 2 It should be noted that Dong Hyuk Kang interestingly highlights intergroup differences in recurrence rates, which could strengthen this study. Therefore, it is suggested to go deeper into this point, with the intention of differentiating this manuscript from what has been previously published.

Answer 2 Thank you for your comment, and we deeply agree with you. In terms of RFS, implementing adjuvant treatment has shown better results than AS, so more discussion is needed to determine which is the best action in the current COVID-19 era, some of which have already been mentioned in 'Discussion'. Seminoma CSI shows a remarkably high treatment success rate even in the case of recurrence, and unnecessary medical burden occurs due to adjuvant treatment, and there is a possibility that side effects may occur due to adjuvant treatment. We should consider that it is more reasonable to regard adjuvant treatment as unreasonable when considering the above factors comprehensively. The contents of this have been reinforced in 'Discussion'.

First, it should be recognized that seminoma CSI shows a notable treatment success rate even in cases of recurrence. Second, there is an unnecessary medical burden due to adjuvant treatment. And lastly, there is a possibility that side effects may occur due to adjuvant treatment. Therefore, even if the disease recurs in the future, it is more reasonable to consider adjuvant treatment as unreasonable considering the reduction in the risk of exposure to COVID-19 and medical ability by implementing adjuvant treatment. [Line 236-242]

Comment 3 On the other hand, the discussion is very short, you should reinforce this section.

Answer 3 Thank you for your comment. We think the discussion was naturally reinforced by answers 1-2.

Comment 4 Finally, there is a pagination mismatch due to the final point in figure 1.

Answer 4 Thank you for your careful comment. It is changed.

Comment 5 In methodology section it is mentioned that "relevant variants" were contemplated, Answer 5 Thank you for your comment. The notation for "relevant variants" is incorrect. It has been modified as follows.

We used search terms such as “seminoma”, “chemotherapy”, “radiotherapy”, “surveillance”, “observation”, and relevant variants. → We used search terms such as “seminoma”, “chemotherapy”, “radiotherapy”, “surveillance”, “observation”, and combinations of these search terms. [Line 76-78]

Round 2

Reviewer 2 Report

Dong Hyuk Kang and colleagues satisfactorily resolved the comments made on the manuscript.

In the discussion section, the strengths of this work were adequately highlighted, with the intention of differentiating it from works previously published by other authors.